# Partisan self-interest is an important driver for people's support for the regulation of targeted political advertising

Katharina Baum[1,2]☯*, Stefan Meissner[3]☯, Hanna Krasnova[1,2]

**1** Weizenbaum Institute for the Networked Society, Berlin, Germany, **2** Faculty of Economic and Social Sciences, University of Potsdam, Potsdam, Germany, **3** Department of Economics, Norwegian School of Economics, Bergen, Norway

☯ These authors contributed equally to this work.
\* katharina.baum@uni-potsdam.de

**Data Availability Statement:** All data files are available from the Open Science Framework database (DOI: 10.17605/OSF.IO/QM7DZ).

**Funding:** This work was partially supported by the Research Council of Norway through its Centres of

## Abstract

The rapid emergence of online targeted political advertising has raised concerns over data privacy and what the government's response should be. This paper tested and confirmed the hypothesis that public attitudes toward stricter regulation of online targeted political advertising are partially motivated by partisan self-interest. We conducted an experiment using an online survey of 1549 Americans who identify as either Democrats or Republicans. Our findings show that Democrats and Republicans believe that online targeted political advertising benefits the opposing party. This belief is based on their conviction that their political opponents are more likely to be mobilized by online targeted political advertising than are supporters of their own party. We exogenously manipulated partisan self-interest considerations of a random subset of participants by truthfully informing them that, in the past, online targeted political advertising has benefited Republicans. Our findings show that Republicans informed about this had less favorable attitudes toward regulation than did their uninformed co-partisans. This suggests that Republicans' attitudes regarding stricter regulation are based not solely on concerns about privacy violations, but also, in part, are caused by beliefs about partisan advantage. The results imply that people are willing to accept violations of their privacy if their preferred party benefits from the use of online targeted political advertising.

## Introduction

The growing popularity of social media platforms has dramatically changed the landscape of political campaigning [1]. Campaigns now increasingly target specific messages to narrow groups of voters on various digital platforms [2–4] (see S1 Text in S1 File for a discussion). Individual-level voter targeting based on data from state-wide voter registries merged with different kinds of public and commercial data has become a widespread practice since the implementation of the Help America Vote Act of 2002 [3, 5–7]. While early efforts of specific targeting largely relied on door-to-door canvassing, telephone calls, and direct mail, today,

Excellence Scheme, FAIR project No 262675. This work has been partially funded by the Federal Ministry of Education and Research of Germany (BMBF) under grant no. 16DII116 and grant no. 16DII127 ("Deutsches Internet-Institut"). This work has been partially funded by the Centre for Ethics and Economics at the Norwegian School of Economics.

**Competing interests:** The authors have declared that no competing interests exist.

platforms like Facebook allow political actors to reach millions of users in a cost-effective way [3, 4, 8]. Thus, spending on online targeted political advertisements has tripled between the 2016 presidential election and the 2020 presidential election in the United States [9].

Observing these astounding levels of targeted political advertising taking place on social media, numerous stakeholders have taken a closer look at these practices and their potentially undesirable consequences for users and society [6, 10–14]. Moreover, private initiatives like ProPublica have raised awareness for online political targeting on Facebook by making political advertisements targeted to specific audiences publicly available [15]. In terms of regulation, there have been several attempts to tighten the election law. For example, in 2017, the Honest Ads Act was proposed, a regulation focused on more rigorous disclosure requirements [16]. Further, some states have introduced legislation that obligates platforms to disclose funding of online targeted political advertisements or to store the advertisements in databases [17]. The Microtargeted Political Ads Act, introduced in 2020, is even more extensive. It would ban online targeted political advertisements based on personal data on online platforms on the federal level [18]. As a reaction to these developments, Twitter, Google, and Facebook have already significantly altered their policies regarding targeted political advertisements [19–22].

A central concern raised by supporters of stricter regulation is a lack of protection and transparency regarding the use of personal data [10–12, 23, 24]. Regulation proponents argue that the sheer amount of available data, advanced predictive modeling, and increasingly sophisticated personalization techniques require new regulatory responses [12]. These demands are aligned with previous research that has established that people value the privacy of their data [25, 26] and that privacy concerns are an important factor in determining attitudes toward the regulation of targeted advertising in general [27] (see S2 Text in S1 File for a discussion). In particular, people seem especially worried about the use of their private data by political actors [28]. Thus, according to public opinion polling, most Americans oppose online targeted political advertisements and consider the use of personal data for online targeted political advertisements to be unacceptable [29, 30].

In this paper, we argue that attitudes toward the regulation of online targeted political advertising are driven not only by concerns over private data misuse. Online targeted political advertising has the potential to influence voting decisions and, as a result, elections [31, 32]. This has consequences for broader societal outcomes, affecting far more than individual data protection. We posit that people take these consequences into account when forming preferences regarding the regulation of online targeted political advertisements. Specifically, we propose that people's preferences regarding the regulation are aligned with their partisan self-interest considerations that reflect one's desire to ensure advantages for one's preferred party and its agenda. Thus, if individuals believe that the opposing party may experience an advantage from targeted political advertisement, they could be motivated by partisan self-interest to support stricter regulation. Indeed, research on public attitudes about other aspects of the electoral process indicates that partisan self-interest is an important factor in people's opinions [33, 34]. Attitudes on gerrymandering, voter ID laws, or same-day voter registration seem to be driven by partisan self-interest [34–36]. Hence, this study seeks to explore whether, in addition to privacy concerns, partisan self-interest is also an important determinant of people's attitudes towards stricter regulation of online targeted political advertising.

To answer this research question, we ran an online survey experiment in the United States, working with a sample of Republican and Democratic participants. First, we established that people regard online targeted political advertisements as opposed to their partisan self-interest. We then manipulated partisan self-interest in our experiment to determine whether there is a causal link between respondents' concerns for partisan advantage and their support for regulation.

## Perceptions about partisan self-interest

It is not immediately obvious whether people perceive online targeted political advertisements to be in line with or opposed to their partisan self-interest. Political parties use targeted political advertisements mainly to mobilize their own partisans who are likely to turnout to vote [3, 37]. Therefore, people's perceptions of whether this advertising benefits or harms their party depend on whether they believe that voters of their party versus voters of the opposing party are mobilized more strongly. In other words, to assess whether the advertisements are to their party's advantage or not, Democrats must guess how Republicans react to mobilizing messages and vice versa. However, campaign messages delivered to targeted recipients remain mostly unavailable to others [15, 32]. Given this limited transparency [24, 31], and the paucity of information about the effects of online targeted political advertising on voters [38], it seems likely that people have difficulties arriving at accurate estimates on the actual impact of online targeted political advertising on others. Therefore, it is plausible that they could hold biased or unfounded beliefs about the issue.

A large body of research documents that people generally believe that others are more influenced by undesirable persuasive mass communication than they are themselves—a phenomenon known as the third-person effect [39, 40] (see S3 Text in S1 File for a discussion). The prevalence of the third-person effect has been documented across a variety of contexts, including press coverage [41] rap music [42], television violence [43, 44], direct-to-consumer advertising [45], media influence [46], social media [47], and, more recently, fake news [48]. Importantly, the third-person effect predicts that people not only believe that others are more influenced by undesirable mass communication (perceptual component), but that these people also take action to rectify the consequences of such persuasive messages (behavioral component) [49]. Frequently, this behavior involves the support for censorship of undesirable media content. For example, this can take the form of censorship endorsement or support for specific restrictions with regard to commercial advertisement [50], violent and misogynous rap lyrics [42], television violence [44] or election campaign news [51].

Notably, past studies have shown that the strength of the third-person effect increases with social distance to the "other" [48, 52, 53]. This finding is important since high levels of polarization and mistrust between Democrats and Republicans in the United States suggest that the social distance between partisans is large [54–60]. Hence, the potential presence of the third-person effect, combined with a large social distance between the parties, suggests that both Republicans and Democrats may believe that opposing partisans are influenced by online targeted political advertising to a larger extent than are supporters of their party. Crucially, this means that the opposing party is perceived as experiencing an advantage from mobilizing messages directed at their respective electorate. Hence, we hypothesize that supporters of both parties will believe that online targeted political advertisements exert a greater influence on partisans of the opposing party than on their co-partisans (H1). This means that online targeted political advertisements are perceived to go against the partisan self-interest of both Republicans and Democrats.

In turn, the perception that opposing partisans are more easily influenced by online targeted political advertisement should result in the desire to impose stricter regulations on these advertisements, as this would mitigate the perceived mobilization advantage of the opposing party. Indeed, past literature on the third-person effect has established that people frequently favor stricter regulation if they believe others to be more influenced by different sorts of media messages [44, 50, 51]. As a result, it can be inferred that people who believe voters of the opposing party are more influenced by online targeted political advertisements than are voters of their own party will also support the regulation of this advertising, perceiving it to be in their

partisan self-interest. Therefore, we further hypothesize that the magnitude of the perceived difference in the effect of online targeted political advertisements on opposing versus fellow partisans is associated with demand for regulation (H2). Correlating privacy concerns to the demand for regulation will also allow us to explore the importance of privacy concerns.

If the demand for regulation is causally explained by partisan self-interest, then altering partisan self-interest considerations of participants should also change their demand for regulation. To test whether it is partisan self-interest that motivates regulation demand, we therefore exogenously manipulated partisan self-interest by changing beliefs about the effect of online targeted political advertising for a randomly selected sample of participants. Informing participants that co-partisans are more susceptible to online targeted political advertisement than supporters of the opposing party should shift these participants' support for regulation downward. Hence, we hypothesize that participants who are informed that supporters of their own party are more influenced by online targeted political advertising are less supportive of regulation than uninformed participants (H3).

This study contributes to the growing literature that links partisan self-interest considerations to attitudes towards election laws [33, 61–65], and adds to the understanding of causality in this relationship [34–36]. Further, our findings contribute to the literature on the third-person effect by providing the first evidence of its existence in the context of online targeted political advertisements. Additionally, we contribute to the scarce literature that supports a causal relationship between third person perceptions and behavior [66, 67]. While most studies in this domain only report correlational evidence, our experiment allows us make causal conclusions. Our results reveal the challenges posed by new technological advances in the political domain and the ensuing need for new regulation. We show that some partisans are willing to oppose regulation if they believe that online targeted political advertising benefits their preferred party, even at the expense of concerns about privacy violations. Our findings further show that attitudes toward regulation are partially driven by biased beliefs about the effect of online targeted political advertising on others, since participants from both parties believe that regulation is in their own partisan self-interest.

## Structure of the study

This study is composed of a correlational and an experimental part. The correlational part provides evidence that participants believe that supporters of the opposing party are more influenced by online targeted political advertising than are supporters of their own party (H1). This way, we establish that supporters of both parties believe that these advertisements are not in line with their partisan self-interest and yield an advantage for the out-party. Importantly, we also show that beliefs about the effect of online targeted political advertising on supporters of the other party relative to supporters of one's own party are positively correlated with a stronger demand for regulation (H2). As a consequence, support for stricter regulation is linked not only to concerns about individuals' privacy, but also to participants' beliefs about partisan self-interest. In the experimental part of the study, we manipulated partisan self-interest by truthfully informing a randomly selected sample of participants that the Republican party benefited more than the Democratic party from the use of online targeted political advertising in the 2016 presidential election. Thereby, we changed Republicans' perceptions of partisan self-interest. Republican recipients of this information were less supportive of regulation than were their co-partisans who did not get this information (H3). This means that Republicans are less in favor of a regulation when they learn that this would go against their partisan self-interest. Thereby, we establish a causal link between considerations of partisan self-interest and people's attitudes toward regulation.

## Experimental design

We conducted an incentivized, between-subjects online survey experiment with a sample of adult Americans identifying either as Democrats or Republicans. The pre-registration of the study is available at the AEA RCT Registry AEARCTR-0005296. The study received an Institutional Review Board (IRB) approval from the Norwegian School of Economics. All participants gave informed consent and the data was collected anonymously. S1 Fig in S1 File provides an overview of the structure of the experiment.

There were three parts to this study, which were all completed within a single session. In the first part, we measured participants' beliefs about the effect of online targeted political advertising on supporters of both the Republican and Democratic parties. In the second part, the experimental manipulation was conducted by informing a random subset of participants about the beneficial effects of online targeted political ads for Republicans. The main dependent variable, participants' attitudes toward the regulation of online targeted political ads, was measured in the third part. Further, we measured posterior beliefs to check whether the treatment group had different beliefs than the control group, and measured respondents' demographics along with a number of other control variables. The following describes measurements and procedures in detail.

To ensure that all participants had the same knowledge on the subject, in the first part of the study, participants were asked to read a text about online targeted political advertising that explained its technical aspects and its typical usage. We then asked participants to consider a hypothetical scenario in which both Republicans and Democrats competed in a close electoral race in which they spent equivalent sums on online targeted political advertising. We elicited participants' beliefs about the extent to which they thought that they personally as well as Republicans and Democrats alike would be influenced by online targeted political advertising, using a five-point scale ranging from "not at all" to "to a very great extent." This measurement corresponds to previous studies on the third-person effect [48, 53]. The order of the questions about Republicans and Democrats was randomized. This measure was used to establish whether participants thought that online targeted political advertisement was in line with their partisan self-interest or not. To address concerns that participants could potentially want to give negative answers about the opposing side while not necessarily believing that such answers had a basis in fact [68, 69], we emulated the approach of previous studies [70], and asked participants to commit to answering the questions to the best of their knowledge.

In the second part of the survey, we manipulated partisan self-interest by manipulating the treatment group's beliefs about who benefits from online targeted political advertisements. To do so, participants were randomly placed in either the treatment or the control group. Participants in the treatment group were informed that controlling for the number of ads people saw, online targeted political advertising on Facebook significantly increased voter turnout for the Republicans in the 2016 presidential election, while having no effect on Democrats. With this wording, we ensured that participants did not look to different levels of campaign spending as a possible cause of the ads' effects. The treatment text was based on results from a working paper that shows that targeted political advertisement on Facebook prior to the 2016 U.S. presidential election increased turnout among Republican, but not among Democratic voters [71].

In the final part of the study, we measured all participants' attitudes towards regulation of online targeted political advertising on a four item, seven-point Likert scale (1 = strongly disagree to 7 = strongly agree), adapted from [48]. The items were: (i) Targeted political advertising should be banned; (ii) I support legislation that requires targeted online political advertising to be clearly marked as targeted; (iii) More regulation is needed when it comes to targeted online political advertising; and (iv) The government is already doing enough to

regulate targeted online political advertising (reverse coded). The order of these responses was randomized. We incentivized honest answers by informing participants that their responses would be sent to the United States Congress in an aggregated and anonymous form [72], stressing that there was no deception in the study.

To determine whether the information treatment succeeded in manipulating beliefs about the effects of online targeted political advertising of participants in the treatment group, all subjects were then asked to make an estimation of the number of interactions (likes, shares, comments) that social media campaigns on Facebook of both Republicans and Democrats received relative to each other prior to the midterm elections in 2018. This enabled us to ascertain whether participants generalized from the treatment information about the 2016 Presidential election and applied it to other elections, and, hence, whether the treatment altered participants' perceptions. We offered a monetary incentive for participants to answer the question to the best of their knowledge [73]. Participants giving the correct answer received a bonus of $1 [69]. In order to control for the possibility that the intervention influenced not only beliefs about online targeted political advertising's persuasiveness, but also about other problematic aspects of such advertising, we also measured whether participants thought the advertising was: (i) socially desirable; (ii) harmful to society (reverse coded); (iii) beneficial to cultural values; and (iv) unfavorable to societal norms (reverse coded) on a ten-point scale.

To assess the level of privacy concerns, we presented participants with a four item, seven-point Likert scale (1 = strongly disagree, 7 = strongly agree) questionnaire (adapted from [74]) in which we asked participants whether they were concerned that their data was: (i) collected and stored by third-parties; (ii) shared with third-parties; (iii) used to display targeted advertising to them; and (iv) used for commercial purposes. The order of the items was randomized. We further included a fifth item as an attention check to ensure that participants carefully read the items. In accordance with our pre-analysis plan, participants who failed this attention check and another attention check were not included in the final sample.

We further collected data for political attitudes in terms of political engagement, subjective political knowledge, participants' level of social and economic conservatism [75], a feelings thermometer towards both the Republican and the Democratic parties [55], and participants' perceived political efficacy [76]. The demographic control variables included age, gender, ethnicity, education, income, household size, use time on the internet, use of an ad-blocker and social media usage.

## Sample characteristics

We collected the data for this survey between January 15, 2020 and January 24, 2020. We collaborated with the survey company Dynata to recruit our participants. For that purpose, we used Dynata's political panel to recruit Republicans and Democrats, as Dynata collaborates with L2. Therefore, we were able to recruit Democrats and Republicans for whom party affiliation was partially verified by their actual voting behavior and partially derived from other known attributes about the participants. In the study, we asked participants for partisanship to further ensure that only Democrats and Republicans participated. That enabled us to avoid recruiting Independents for our study. In total, we recruited a sample of 1549 American participants with quotas on age, gender, region and party affiliation. The distribution of age, gender and region broadly followed the overall distribution in the general population. In accordance with our pre-analysis plan, the fastest 3% of respondents were removed from the sample to increase data quality. On average, participants were 47.49 (SD = 16.48) years old. Of the sample, 50.55% were female and 25.05% were non-white. The participants were better educated

than the overall population of the United States. S1 Table in S1 File provides an overview of the characteristics of our sample.

Among the participants, 777 identified as Republicans and 772 as Democrats. Given the nature of the experimental design, Independents were not included in the study. We randomly assigned the participants to either the treatment group (755 participants: 369 Democrats, 386 Republicans) or the control group (794 participants: 403 Democrats, 391 Republicans). Treatment assignment was balanced taking into consideration observable characteristics and pre-treatment beliefs (S2 Table in S1 File).

## Results

This section presents the results of the study. First, we will present evidence supporting the hypothesis that supporters of both parties believe that supporters of the opposing party are influenced to a larger extent by online targeted political advertising than are supporters of their own party (H1). This implies that they believe that the use of online targeted political advertising undermines their partisan self-interest. We will then present correlational results regarding the link between these beliefs, privacy concerns and support for stricter regulation (H2). Last, we will present our findings about the causal role of beliefs about the effects of online targeted political advertising on attitudes towards regulation (H3). The analysis was performed using Stata SE 16.0. The data, full instructions for participants, analysis code and variable coding are available at 10.17605/OSF.IO/QM7DZ.

### Beliefs about the differential effect of online targeted political advertising on opposing versus fellow partisans

This section reports results for Hypothesis 1. Fig 1 shows the participants' beliefs about the extent to which online targeted political advertising influences Republicans and Democrats. We found that Republicans believed that Democrats ($\mu = 3.20$, SD = 1.18) were significantly more influenced than Republicans ($\mu = 2.83$, SD = 1.10, Wilcoxon-signed-rank-test, z = -8.67, p < 0.001, r = 0.41). In contrast, Democrats stated that they believed that Republicans ($\mu = 3.41$, SD = 1.17) were more influenced than were Democrats ($\mu = 2.94$, SD = 1.02, Wilcoxon-signed-rank-test, z = -11.336, p < 0.001, r = 0.31). This result supports our Hypothesis 1 that claimed that Republicans as well as Democrats expressed the belief that supporters of the opposing party are more influenced by online targeted political advertisement than are supporters of their own party. This finding is consistent with previous literature on the third-person effect [39, 40]. The magnitude of this perceived difference in the effect of online targeted political advertisement on opposing party supporters relative to supporters of their own party is not significantly different between Republicans and Democrats (two-sided Welch t-test, t (1540) = 1.61, Cohen's-d = 0.08, p = 0.11). Believing that voters of the opposing party are influenced to a larger extent by these advertisements than voters of one's own party, hence being more easily persuaded to vote for their respective parties, indicates that both Republican as well as Democratic think the other party gains more votes by using these ads. This implies that participants perceive such advertising as harmful to their own party, undermining their partisan self-interest. Further, we found that participants believed that online targeted political advertising had a smaller influence on themselves ($\mu = 2.39$, SD = 1.21) than on others. The perceived desirability of these advertisements was slightly lower than medium ($\mu = 4.66$, SD = 2.01, measured on a ten-point scale).

Exploring the size of the belief gap between the perceived effect that online targeted political advertising has on supporters of the other versus one's own party shows that it is correlated to a number of different attitudes that participants hold (see Table 1). We found that affective

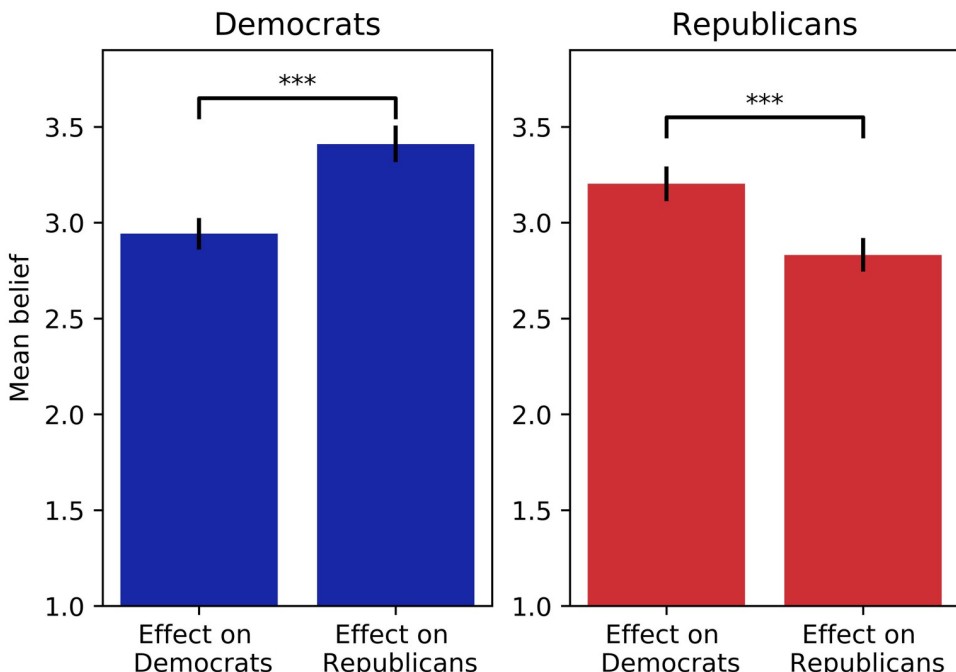

**Fig 1. Beliefs about the effect of online targeted political advertising.** Participants' beliefs about the effect of online targeted political advertising on Democrats and Republicans. Beliefs are measured on a five-point Likert scale (1 = "not at all", 5 = "to a very great extent"). The bars show 95% confidence intervals. $^{*}$ $p < 0.05$, $^{**}$ $p < 0.01$, $^{***}$ $p < 0.001$.

**Table 1. Regression of determinants predicting the size of the difference between the perceived effect of online targeted political advertisement on the other party versus one's own party.**

| | Effect on other minus effect on own party | | | |
|---|---|---|---|---|
| | Coef. | Robust SE | p-value | 95% CI |
| Affective polarization | 0.170 | 0.032 | <0.001 | 0.107, 0.232 |
| Ideological polarization | 0.112 | 0.031 | <0.001 | 0.051, 0.174 |
| Desirability of advertising | -0.149 | 0.017 | <0.001 | -0.183, -0.116 |
| High political knowledge | 0.133 | 0.065 | 0.040 | 0.006, 0.261 |
| Use of internet in hours | 0.003 | 0.005 | 0.642 | -0.008, 0.132 |
| Use of ad-block | 0.018 | 0.029 | 0.530 | -0.039, 0.076 |
| User of social media | 0.000 | 0.085 | 0.995 | -0.166, 0.167 |
| Attitude towards government regulation | -0.013 | 0.016 | 0.424 | -0.046, 0.019 |
| External efficacy | -0.002 | 0.001 | 0.119 | -0.004, 0.001 |
| Politically active | 0.078 | 0.060 | 0.193 | -0.039, 0.197 |
| Constant | 0.808 | 0.197 | <0.001 | 0.422, 1.193 |
| Demographics | Yes | | | |
| Observations | 1464 | | | |
| $R^2$ | 0.148 | | | |

Note: The table reports the results for an OLS-regression with the difference between how much participants thought online targeted political advertising influences voters of the other party minus how much they thought it influences voters of their own party as dependent variable. The dependent variable is standardized. Affective and ideological polarization are standardized. User of social media is a dummy for the use of social media, use of ad-block is a dummy for ad-block use. Political engagement is a dummy variable for being politically active within the last year, political knowledge is a dummy for above median knowledge. Demographics include age, gender, income, education, ethnicity, and household size. S3 Table in S1 File provides an overview of all variables in the regression.

and ideological polarization, perceived desirability of the advertising, and high subjective political knowledge are significant predictors of this gap. Participants holding a more negative view of the opposing party as measured on a feelings thermometer (i.e. affective polarization) showed a larger belief gap ($\beta = 0.170$, $p < 0.001$). We also found that the level of conservatism for Republicans and liberalism for Democrats (i.e. ideological polarization) as measured on a scale for social and economic conservatism [75] positively predicted the size of the belief gap ($\beta = 0.112$ $p < 0.001$). Participants who saw the advertising as more socially and culturally desirable reported a significantly smaller gap in beliefs between their own party and the other party ($\beta = -0.149$, $p < 0.001$). Taken together, these results suggest that people's belief that supporters of the opposing party are more influenced than supporters of their own party by online targeted political advertising is linked to a negative perception of the opposition and a more general dislike of online targeted political advertising. This conclusion accords with previous literature on the third-person effect that suggests that people's belief about the influence of media messages on others relative to themselves correlates with the social distance to the other and a negative perception of the message [48, 52, 53]. Moreover, participants who self-reported a high level of political knowledge reported a larger gap between their own party and the other party ($\beta = 0.133$, $p = 0.04$).

## The relationship between beliefs about voters' susceptibility to online targeted political advertisement and support for its regulation

In this section, overall demand for regulation as well as results on Hypothesis 2, namely the association between beliefs about the effect of online targeted political advertisement and support for its regulation are presented.

When analyzing both the treatment and the control group together, participants were slightly in favor of regulation of online targeted political advertisement on average ($\mu = 4.82$, SD = 1.18, Cronbach's-$\alpha = 0.67$). Overall, 70% of all participants supported stricter regulation. In the control condition, support for stricter government regulation was higher among participants who identified as Democrats ($\mu = 5.06$, SD = 1.10) compared to Republicans ($\mu = 4.59$, SD = 1.21, two-sided Welch t-test, t(782) = 5.79, Cohen's-d = 0.41, $p < 0.001$). We further found that, on average, participants in both conditions were concerned about the use of their personal data in online targeted political advertising ($\mu = 5.63$, SD = 1.25, Cronbach's-$\alpha = 0.90$). This concern was not significantly different (two sided Welch t-test, t(1529) = 0.10, Cohen's-d = 0.05, $p = 0.31$) between Democrats ($\mu = 5.67$, SD = 1.26) and Republicans ($\mu = 5.60$, SD = 1.25). S10 and S11 Figs in S1 File show the distributions of support for regulation and privacy concerns.

To investigate the relationship between participants' beliefs about the influence of online targeted political advertisement and support for its regulation an OLS-regression was performed, analyzing participants in the control group only (see Table 2). This was done in order shed light on the association between these variables without incorporating the effect of the treatment manipulation. Our results show that among control group participants, beliefs about the influence of online targeted political advertisement on voters of the other party relative to its perceived influence on voters of their own party is a significant predictor of support for stricter regulation of such advertisement (belief other party—own party, $\beta = 0.124$, $p < 0.001$). This confirms Hypothesis 2, which claimed that the more people think that members of the out-party are more susceptible to online targeted political advertising than members of their in-party, the more they are in favor of regulating such advertisement. Furthermore, privacy concerns were significant predictors of participants' support for regulation in the control condition ($\beta = 0.257$, $p < 0.001$). We find no significant link between

**Table 2. Regression of determinants predicting the willingness to support stricter regulation of online targeted political advertising, control group.**

| | Support for regulation | | | |
|---|---|---|---|---|
| | Coef. | Robust SE | p-value | 95% CI |
| Belief other party—own party | 0.124 | 0.035 | <0.001 | 0.055, 0.193 |
| Belief about the effect on oneself | 0.052 | 0.039 | 0.187 | -0.025, 0.129 |
| Privacy concerns | 0.257 | 0.045 | <0.001 | 0.169, 0.344 |
| Observations | 754 | | | |
| $R^2$ | 0.125 | | | |
| Demographics | Yes | | | |
| Social Media use | Yes | | | |
| Political Engagement | Yes | | | |

Note: The regression only includes participants in the control group who answered all questions of the survey. The table reports results from an OLS-regression with respondents' support for stricter regulation of online targeted political advertisement as the dependent variable. Belief other party—own party is defined as the difference in participants' beliefs about the effect that online targeted political advertising has on the other party minus its effect on the own party. Belief about self is people's belief about the effect that online targeted political advertising has on themselves. Privacy concerns are measured on a seven-point Likert scale (1 = strongly disagree, 7 = strongly agree). All variables were standardized. Demographic information includes age, education, income, household size, gender, and ethnicity. Social media use includes whether the participant uses social media, the time they spent online in general (in hours), and and the use of an ad-blocker. Political engagement measures include a variable for being politically active within the last year, external political efficacy, political knowledge, and attitudes towards government regulation in general. S11 Table in S1 File provides an overview of all variables in the regression.

participants' beliefs about the effect that online targeted political advertising has on themselves and their support for stricter regulation (belief about the effect on self, $\beta = 0.052$, p = 0.19).

To further investigate our findings, we also ran an OLS-regression using participants' beliefs about the effect of online targeted political advertising on the opposing party and their beliefs about the effect on their own party as individual independent variables. The more participants thought that members of the opposing party would be influenced by online targeted political advertisement, the more they supported stricter regulation of such ads ($\beta = 0.169$, p < 0.001). Their belief about the effect of these ads on voters of their own party was negatively correlated to support for regulation, but not significantly ($\beta = -0.043$, p = 0.26) (see S4 Table in S1 File).

## The causal effect of beliefs about voters' susceptibility to online targeted political advertisement and support for its regulation

This section reports the experimental results, which were predicted by Hypothesis 3. In the treatment condition, we manipulated partisan self-interest by informing a randomly selected subgroup of Republicans and Democrats that the Republican party benefited more from the use of online targeted political advertising in the 2016 presidential election than did Democrats. To determine whether this information shifted respondents' support for stricter regulation of online targeted political advertising, we compared levels of support for regulation between Democrats and Republicans in the treatment and the control group. With Republicans, we found significantly lower support for stricter regulation of online targeted political advertising in the treatment than in the control group (two-sided Welch t-test, t(776) = 2.08, Cohen's d = 0.15, p = 0.04). This confirms Hypothesis 3, showing that Republicans who learn about the advantageous effects of online targeted political advertising for their party are less in favor of regulation than their uninformed co-partisans. This means that Republicans are less in favor of a regulation that goes against their partisan self-interest. These effects remained qualitatively the same when examining only participants who wanted their opinions to be

considered by Congress (98.7% of the sample) and participants who expressed trust in the information that they had received about the effect of online targeted political advertisement (85.7% of the treatment group), although in the latter case, the effect became insignificant for Republicans (S5 and S6 Tables in S1 File).

For Democrats, we found no difference in levels of support for regulation between the treatment and the control group (two-sided Welch t-test, t(759) = -0.55, Cohen's d = 0.04, p = 0.58). This result is in accordance with the finding that Democrats believed Republicans are more influenced by political online advertising than members of their own party. Therefore, the information we gave them corresponded with their pre-existing beliefs, and should not alter their regulation demand. S15 and S16 Figs in S1 File show the distribution of answers for Democrats and Republicans in the treatment and the control groups.

To make sure the shift in demand for regulation resulted from a shift in beliefs about the extent to which online targeted political advertisement influenced voters of each party, we tested whether the treatment group had different beliefs regarding Republicans' susceptibility to online targeted political advertisements. This was done by measuring beliefs about the effect of online targeted political advertising on Republicans and Democrats a second time after the treatment information, this time in the form of beliefs over interactions on social media posts. Fig 2 shows the effect that the treatment information had on beliefs about social media interactions in the 2018 midterm election. We found that in this incentivized question, Republicans and Democrats in the control condition reported beliefs that were qualitatively similar to the first measure of their stated beliefs. Uninformed Republicans believed that Democrats received more interactions in the run-up to the 2018 midterm elections while uninformed Democrats believed that Republicans received more interactions. Responses to this question and to the pre-treatment measure of the effects of online targeted political advertising on Republicans and Democrats are well correlated (r = 0.24, p < 0.001, see S12 Fig in S1 File). However, results differed for respondents in the treatment condition: Republicans who received the treatment information about online targeted political advertisements helping them in the 2016 Presidential election reported that they believed that Republicans received more interactions in 2018. This result represents a significant divergence in beliefs between informed and uninformed Republicans that corresponds to the information that they received, $\chi^2$(13.04, N = 771), p = 0.04. For Democrats, though, no shift in their beliefs about the 2018 midterm elections was detected, $\chi^2$(6, N = 764), p = 0.65. This further corroborates the finding that the treatment information altered partisan self-interest considerations and caused Republicans to demand less regulation due to a shift in beliefs about the advantageous effect of online targeted political advertisement for their party. Moreover, this result shows that the treatment information reinforced Democrats' pre-existing beliefs about Republicans being more susceptible to online targeted political advertisement. Therefore, their regulation demand is not altered by our treatment.

To preclude the possibility that the information about the effect of online targeted political advertising changed participants' perception of how desirable such advertising is or participants' privacy concerns, we tested for significant differences in these measures between the treatment and the control group. We found that, in general, participants viewed the desirability of using online targeted political advertising as slightly lower than medium ($\mu$ = 4.66, SD = 2.01). Comparing the ratings of the desirability of online targeted political advertising for Republicans in the treatment ($\mu$ = 4.75, SD = 2.00) and the control group ($\mu$ = 4.85, SD = 2.00), we found no statistically significant difference (two-sided Welch t-test, t(769) = 0.69, Cohen's d = 0.05, p = 0.49). The same result was found for Democrats in the treatment ($\mu$ = 4.42, SD = 2.03) and the control group ($\mu$ = 4.61,SD = 2.00 two-sided Welch t-test, t(755) = 1.28, Cohen's d = 0.09, p = 0.20). We also found no significant differences in privacy concerns

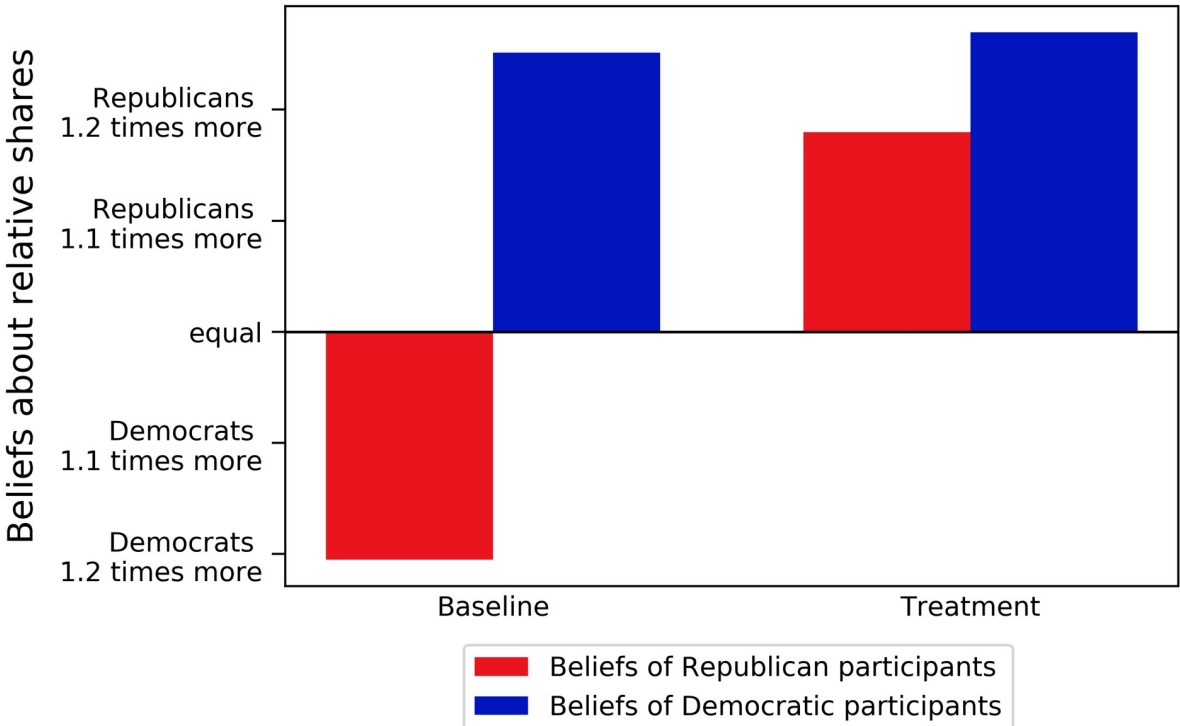

**Fig 2. Beliefs about social media engagement in the 2018 midterm elections.** Participants' beliefs about the ratio of interactions in the run-up to the 2018 Midterm election. This was measured on a scale that ran from "Democrats three times more than Republicans" to "Republicans three times more than Democrats" with "Equal" as the mid-point.

between the treatment and the control groups, for both Democrats (two-sided Welch t-test, t(759) = 0.10, Cohen's d = 0.05, p = 0.46) and Republicans (two-sided Welch t-test, t(768) = -0.70, Cohen's d = -0.05, p = 0.49). Hence, the treatment manipulation altered Republicans perceptions about how much other voters are influenced by online targeted political advertising, while neither affecting perceived desirability nor privacy concerns.

Exploratory data analysis reveals that the effect of the information on Republicans was heterogeneous between different levels of conservatism (see S17 and S18 Figs in S1 File). We found that for those Republicans scoring below the median in social and economic conservatism, the information that their party benefited from the use of online targeted political advertisement did not significantly change their support for regulation compared to the same group who did not receive this information (two-sided Welch t-test, t(403) = 0.16, Cohen's d = 0.02, p = 0.87). The support for stricter regulation of online targeted political advertisement among Republicans scoring at or above the median in economic and social conservatism differed significantly between the treatment and the control groups (two-sided Welch t-test, t(373) = 2.59, Cohen's d = 0.27, p = 0.01). This effect cannot be attributed to initial differences in the support for regulation in the baseline condition between above median and below median conservative Republicans (two-sided Welch t-test, t(347) = 0.47, Cohen's d = 0.05, p = 0.64).

## Discussion

The present study reveals that both Republican and Democratic participants in our sample believe that supporters of the opposing party are influenced by online targeted political advertising to a greater extent than are supporters of their own party. In the context of our study,

this implies that both Democrats and Republicans assume that the opposing party benefits more from such advertisement, as they think their members are more easily persuaded by being exposed to online targeted political advertisement. Hence, both Democrats and Republicans regard online targeted political advertisement as disadvantaging their partisan self-interest. We further found that both this belief and people's concern over data privacy significantly predict people's support for policies limiting the use of such ads. This indicates that people are in favor of regulation since they seek to protect their private data, but also due to an urge to mitigate the perceived advantageous effect of the advertisements for the opposing party. These results therefore provide evidence that the support for stricter regulation of online targeted political advertising is partially motivated by partisan self-interest. Causal evidence that substantiates these findings was provided in the experimental part of the study. The results of the experimental manipulation of partisan self-interest show that Republicans who were informed about the beneficial effects of online targeted political advertisement for their own party reported lower support for regulation than did Republicans in the control group. Therefore, we are able to show that the perception bias is causally linked to Republicans' support for stricter government regulation. This suggests that participants might make a trade-off in favor of partisan self-interest and contrary to concerns about the violation of data privacy. We found that this effect is not present with all Republican participants, but is concentrated among those with the highest levels of conservatism. This finding concords with the idea that people trade-off personal costs, such as privacy concerns, against partisan self-interest. As more conservative Republicans gain more from an electoral advantage for their party, they are more willing to accept violations of privacy if these violations provide their preferred party with a benefit in an election.

These results contribute to the findings of previous research examining motivations behind attitudes toward election laws. While multiple studies provide evidence that partisan self-interest considerations are associated with attitudes towards the election law, many of these studies are cross-sectional and, hence, do not establish causality [33, 61–64, 77]. Several recent studies use experimental designs to measure causal effects of partisan self-interest on attitudes toward the election law [34–36]. We follow this line of research to provide more experimental evidence of the causal relationship between partisans' self-interest considerations and their opinions towards the election law. Our findings support the idea that the broader public favors regulation based on their partisan self-interest, and supports laws that contribute to the electoral success of their preferred party.

Our findings further add to an emerging body of literature that shows that some people are willing to make trade-offs between established democratic norms and partisan self-interest [78–81]. According to our results, participants holding the strongest policy views have the greatest reaction to the information that online targeted political advertising benefits their party. This finding accords with previous findings that people are willing to accept the undermining of democratic or moral principles if it benefits their political goals. For example, previous research has documented that partisans are less willing to take corrective measures against politically biased messages if they benefit their party [82].

Similarly, in our case, people's attitudes towards the regulation of online targeted political advertising are partially driven by the desire to set rules that benefit people's preferred party, even if they view online targeted political advertising as harmful to societal norms. This behavior might be perceived as a threat to perceptions of the fairness of elections, which could then undermine peoples' support for a electoral system that relies on a shared understanding of democratic norms [83–88]. We show that the rise of new technologies could potentially contribute to perceptions of "democratic backsliding" [78], as people might be willing to use the newly-required technologies to pursue their partisan self-interest. We further show that beliefs

about the impact that new technologies have on the electoral process are crucial to our understanding of public attitudes towards them. This finding contributes to a wider body of literature that investigates how potentially erroneous beliefs that people hold drive their behavior [89–95].

This study reveals that it is difficult to understand public preferences for certain policy measures without understanding the beliefs that people hold about key variables that are affected by these policies. Preferences for regulation of online targeted political advertising are currently driven in part by third-person perceptions, leading to biased beliefs about their effect. This situation could lead to potentially sub-optimal policy decisions, as politicians might follow public preferences that are driven by biased beliefs. Our findings underscore the necessity of providing the public with truthful information about the effect of online targeted political advertising. We show that support for stricter regulation among Republicans would be significantly lower if they were correctly informed about the effect that it had on the 2016 Presidential election, because they underestimate the positive effect that online targeted political advertising might have had or will have on their own party.

Previous research on the third-person effect found evidence for a gap between the perceived effect of persuasive mass communication on the self and on others [39, 40]. Furthermore, correlational research supports the hypothesis that this gap motivates people in performing mitigating actions against the negative consequences of such persuasive communication [49]. Our study adds to this literature in three ways. First, this study is the first to show that a perceptual gap exists in the context of targeted online political advertising. Second, this study establishes a causal link between the perceptual gap described by the third-person effect and a behavioral measure for support for government regulation. Thereby, we add to the scarce previous studies that show a causal relationship between third person perceptions and behavior [66, 67]. By manipulating the perception gap of Republicans in our information treatment downward, and by showing that this decreases their support of the mitigating action, we were able to show causality between perception and behavior. Third, our results also add to previous studies reporting that the third-person perception increases with social distance, or between in-groups and out-groups [48, 52, 53]. To the best of our knowledge, this is the first study to show that the gap between Democrats and Republicans in their perceptions of the influence of undesirable mass communication is strongly linked to affective as well as ideological polarization, and it is the first study to measure this outcome with an unincentivized and an incentivized measure.

Our results have several limitations. We were unable to show similar causal results for Democratic supporters. We found a strong correlation between the beliefs that Democrats report about the effect that online targeted political advertising has on Republicans and their support for stricter government regulation, but cannot claim causality for this group. Due to our incentivization of our outcome measure, we needed to truthfully inform participants that we were not using deception in this study, and we were therefore unable to manipulate Democrats' beliefs in a way that was equivalent to that used with Republicans. At the time of designing the study, no scientific evidence was available to support the claim that Democrats benefited more from online targeted political advertisement in some election. Future research should address this shortcoming and examine possible partisan self-interest motives of Democrats in this context more closely. We have not used Independents as a control group due to the mixed political leanings of Independents. Only a small percentage of Independents do not favor either the Democrats or the Republicans [96]. Hence, their attitudes would likely depend on the composition of the sample. In addition, our treatment manipulation stated that online targeted political advertisements "significantly increased the number of votes for the Republican party, but not for the Democratic party. Hence, online targeted political ads influenced Republican voters, but did not influence Democratic voters." This information is based on

results from a study on the effect of online targeted political advertisements on Facebook during the 2016 U.S. presidential campaign on voter turnout and candidate choice [71]. These results were the only results about the influence of online targeted political advertisements on voters available to us at the time of designing the experiment. While the results of that research show that overall, these ads did not impact voting behavior of Democrats, the phrasing that Democrats were not influenced at all, as stated in our treatment, might not resemble actual beliefs of Democrats or Republicans. Hence, while we are able to show that partisan self-interest considerations motivated a shift in regulation demand for Republicans, the size of that effect might be smaller outside of this study's setting. After our survey had been in the field, the authors of the 2018 working paper published a newer version, in which some of the earlier results were revised. In the new version, the authors conclude that online targeted political advertisements increased turnout for the Republican party, and decreased turnout for the Democratic party [97]. Our treatment text does not incorporate these latest results. Further, the main measure of interest, participants' support for stricter government regulation, indicates relatively low-scale reliability (Cronbach's $\alpha = 0.67$). However, exploratory results show that a reduced scale (excluding the fourth item) has higher reliability (Cronbach's $\alpha = 0.75$) and that all of our main results are robust to the reduced scale (see S7 and S8 Tables in S1 File). Moreover, to address concerns that participants might be principally opposed to the idea of banning targeted advertising, we excluded the first item that asks for support for a ban. The results of this analysis are available in S9 and S10 Tables in S1 File. These findings replicate the effect we report in the results section of this paper. Another limitation is that even though participants read an explanatory text about the functionalities and use cases of online targeted political advertising in the beginning of the survey, we could not rule out the possibility that participants had diverging levels of knowledge of the topic. While this does not limit the interpretation of our treatment manipulation, it could be an important predictor of participants' estimates of the influence of online targeted political advertisements on others. Future studies should address this by measuring participants' knowledge after they have read an explanatory text.

This paper develops a new experimental paradigm to study people's attitudes towards technological change which has an influence on elections. We show that support for or opposition to the regulation of new technology that has implications for the political process is driven by potentially biased beliefs about how the use of this technology affects political outcomes for one's preferred party. Therefore, our findings add to a growing policy debate and underscore the necessity of making the effects of online targeted political advertising transparent and of truthfully informing the public about the effects of the new technology so that the public can fully and knowledgeably realize their true attitudes. We believe that more research is necessary to fully understand the public's attitude towards these innovations, especially regarding beliefs about the spread and effect of false information and divisive messages. Further, our result indicating that people consider the broader societal effects of targeted political advertising might have implications for certain aspects of targeted commercial advertising. We would encourage future research to investigate whether similar mechanisms would motivate people to oppose, for example, the use of targeted advertising to promote socially undesirable consumption, such as smoking, drinking or other unhealthy behavior.

## Supporting information

**S1 File.**
(PDF)

## Acknowledgments

We thank Alexander W. Cappelen, Björn Bartling, Erik Ø. Sørensen, George Loewenstein, Alex Imas, seminar participants at the Weizenbaum Institute and the Norwegian School of Economics as well as the editor and anonymous reviewers for their helpful feedback and comments.

## Author Contributions

**Conceptualization:** Katharina Baum, Stefan Meissner, Hanna Krasnova.

**Formal analysis:** Katharina Baum, Stefan Meissner.

**Methodology:** Katharina Baum, Stefan Meissner, Hanna Krasnova.

**Supervision:** Hanna Krasnova.

**Writing – original draft:** Katharina Baum, Stefan Meissner.

**Writing – review & editing:** Katharina Baum, Stefan Meissner, Hanna Krasnova.

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
