## [Decision Letter · Decision Letter 0]

16 Dec 2020

PONE-D-20-33004

Partisan self-interest is an important driver for people's support for the regulation of targeted political advertising

PLOS ONE

Dear Dr. Baum,

Thank you for submitting your manuscript to PLOS ONE. After careful consideration, we feel that it has merit but does not fully meet PLOS ONE’s publication criteria as it currently stands. Therefore, we invite you to submit a revised version of the manuscript that addresses the points raised during the review process.

Both reviewers and I see potential in your manuscript, but the reviewers have raised several concerns. I will not reiterate the points, as the reviewers have provided detailed comments. Please address all reviewer comments in a revision.

We look forward to receiving your revised manuscript.

Kind regards,

Natalie J. Shook

Academic Editor

PLOS ONE

Journal Requirements:

2. During our internal checks, the in-house editorial staff noted that you conducted research or obtained samples in another country. Please check the relevant national regulations and laws applying to foreign researchers and state whether you obtained the required permits and approvals. Please address this in your ethics statement in both the manuscript and submission information.

Reviewers' comments:

Reviewer's Responses to Questions

**Comments to the Author**

1. Is the manuscript technically sound, and do the data support the conclusions?

Reviewer #1: Yes

Reviewer #2: Partly

2. Has the statistical analysis been performed appropriately and rigorously? 

Reviewer #1: Yes

Reviewer #2: I Don't Know

3. Have the authors made all data underlying the findings in their manuscript fully available?

Reviewer #1: Yes

Reviewer #2: Yes

4. Is the manuscript presented in an intelligible fashion and written in standard English?

Reviewer #1: Yes

Reviewer #2: Yes

5. Review Comments to the Author

Reviewer #1: Authors present a timely and important study, and show that partisan self-interest is an important factor in support for regulation of political targeting. I want to applaud the transparency efforts of the authors. I am also impressed by the authors’ rigor. However, I do have some major concerns that I will now list below.

1. I was surprised the authors took the US as a case to study attitudes toward regulation of targeted political advertising. I understand that the US is a very polarized nation, so in that sense the US is a natural case. However, online political advertising is political speech and thus protected by the first amendment. As such, any effort to regulate targeted online advertising will be challenging at best. Looking at the four items that together measure attitude toward regulation, especially the first item: “Targeted political online advertising should be banned” seems to be measuring non-attitudes. I would invite the authors to construct a new scale with only items 2, 3, and 4 and subsequently run the relevant analyses once more. This scale will better capture US citizens’ attitudes toward regulation.

2. I think that knowledge about political targeting could be an important moderator for partisans’ estimates of the influence of targeted advertising on others. Why did the authors not include this variable and would the authors agree that this omission limits the interpretability of the findings?

3. I am concerned about the phrasing of the treatment. Authors phrased the treatment as follows: “[…] The researchers found that the targeted political ads significantly increased the number of votes for the Republican party, but not for the Democratic party. Hence, targeted political ads influenced Republican voters, but did not influence Democratic voters.”

First, this treatment text is not ‘truthfully’ informing participants about the effects in 2016. Essentially this text states that zero democratic voters were influenced? This can’t be the case. A small consequence Is that participants were, in fact, deceived.

Second, and more importantly, why phrase the treatment in such an extreme, zero-sum, way? Why not state that targeting was thought to have benefitted republicans more than democrats? This treatment now is very artificial, unrealistic even. Which is a shame because there was a perfectly natural way to do it. As a result I wonder what the experimental effect really means? Of course republicans are not going to oppose something that gives them, and only them, an advantage. Such an advantage is worth setting aside, say, privacy concerns because it gives them back electoral victory and thus important policies. But in reality, the republican advantage, in 2016, was much more modest than described in the treatment. In a more realistic scenario republican partisan self-interest might have played a (much) smaller role.

4. Would it not be better to include independents as a control group? Or to compare the scores to a ‘balanced condition’, where both democrats as well as republicans benefit about equally from targeting?

5. Authors did never explicitly measure partisan self-interest. This makes it difficult to show the hypothesized mechanism. Why did they not measure this key concept?

6. On p. 11 the authors state that perceived desirability of online targeted ads is low. But on a ten-point scale, perceived desirability is 4.66, sd = 2.01. I would not call this low.

I believe this study makes an important contribution to the field of online political advertising research, especially when issues 1 and 3 are addressed.

Reviewer #2: Manuscript PONE-D-20-33004 evaluates if Americans’ support for hypothetical regulations of microtargeting change when subjects are informed that microtargeting benefits Republicans while having no effect on Democrats. Results are not surprising based on the many published studies that find support for various (proposed) election laws shifts when partisans are informed that the proposed law disproportionately helps/hurt their party. In this experiment, the effects were only significant for Republicans – though, this is not clearly represented in the current wording of the abstract. I have a few concerns about the experimental components of the manuscript, as well as the article’s framings. Order is consistent with the manuscript’s layout and does not indicate priority.

Microtargeting in the United States is very different than microtargeting in Europe due to the accessibility of publicly available electronic lists of registered voters. States are required by the Help America Vote Act of 2002 (HAVA) to compile and maintain these lists that include basic information (e.g. voter history, party registration, and race/ethnicity). This information is far more valuable for campaigns than data they can purchase or acquire elsewhere. The authors do not seem to understand the distinctions between microtargeting in the United States and elsewhere. And they fail to cite any scholars of US microtargeting (e.g. Hersch, Hillygus, Endres, Nickerson, Rogers, Panagopoulos).

The study’s motivation feels like a straw man argument. The authors note on page 3, “a heated public debate calling for stricter regulations has accompanied the emergence of such ads.” However, none of these citations reference a “heated public debate” about regulating microtargeting in the United States, and I am not aware of a debate or proposed legislation in any US state or nationally. The merits of microtargeting are more frequently debated outside the US than inside. After all, the United States Congress paved the way for political microtargeting through their HAVA legislation (see Hersh 2015). The authors appear to be conflating debates about political misinformation, dark money in politics, and false statements on social media with political microtargeting. Further, the authors are somewhat cavalier with their references throughout the manuscript.

I do not consider the treatment, as described in the text, to be truthful. Though, the exact treatment language is not quoted in the text. The authors note, “we truthfully informed a randomly selected sample of participants that the Republican party benefited more than the Democratic party from the use of targeted political advertising in the 2016 presidential election” (page 6). I am not aware of any legitimate, peer reviewed publication that support this claim. It is possible, that Republicans benefitted more than the Democrats in terms of vote choice / persuasion, fundraising, (de)mobilization, but one CANNOT simply assume that the party of the winning candidate benefitted more than the party of the losing candidate. Later in the paper the treatment is described slightly differently as, “Participants in the treatment group were informed that controlling for the number of ads people saw, targeted political advertising on Facebook significantly increased voter turnout for the Republicans in the 2016 presidential election, while having no effect on Democrats ” (Page 8). The working paper cited to support this claim does not support it. In fact, it argues that targeted advertising had a negative effect on the turnout of Democrats.

More information is needed about the survey. Was there any quota sampling? If so, which variables were used? What was the time interval between each survey wave? What was the attrition rate? Some aspects/concepts are incorrectly described (e.g. a response scale from "not at all" to "to a very great extent" is NOT a Likert scale). Do you have a citations for the description of L2 as “the largest voting tracking service in the United States.” How did L2 and/or Dynata identify Democrats and Republican? Is this based on self-reports from an earlier survey? Party registration (not all states have party registration)? Votes in primary elections (most registered voters do not vote in primaries)?

The "manipulation check", “all subjects were then asked to make an estimation of the number of interactions (likes, shares, comments) that social media campaigns on Facebook of both Republicans and Democrats received relative to each other prior to the midterm elections in 2018” does not appear to directly evaluate whether the treatment group read and/or processed the information in the treatment.

Finally, all control variables should be collected pre-treatment (See Gerber and Green 2012; Montgomery, Nyhan, Torres 2016; and many others), but it appears many were collected in wave 3 after the treatment was delivered (wave 2).

6. PLOS authors have the option to publish the peer review history of their article (what does this mean?). If published, this will include your full peer review and any attached files.

Reviewer #1: No

Reviewer #2: No

---

## [Author Response · Author response to Decision Letter 0]

1 Mar 2021

Dear Editor,

Dear Reviewers, 

We would like to thank you all for your very constructive and helpful feedback on our submitted manuscript and for your general interest in the topic. We highly appreciate that you have taken the time and effort to provide us with these constructive and insightful suggestions to improve the quality of our manuscript. Based on your suggestions we revised our manuscript and tried to incorporate all of the points raised. As suggested, we especially focused on rewriting the introduction to incorporate more research by scholars of targeted political advertisement and to revise our main argument. Based on your comments, we now also explain in more detail the phrasing of our treatment text and how we manipulated partisan self-interest. Further, we excluded control variables that were measured after the treatment from our analysis. In addition to this, we have also worked on strengthening the theory behind our hypotheses and making our reasoning more evident. Please find our detailed responses to your comments in the following section.

Comments from Reviewer 1

Authors present a timely and important study, and show that partisan self-interest is an important factor in support for regulation of political targeting. I want to applaud the transparency efforts of the authors. I am also impressed by the authors’ rigor. However, I do have some major concerns that I will now list below.

Response: Dear Reviewer, we would like to thank you for the appreciation of our submitted manuscript and thank you once more for taking the time and effort to provide these very constructive and insightful suggestions. We hope that we addressed them appropriately in our revised manuscript. 

1. I was surprised the authors took the US as a case to study attitudes toward regulation of targeted political advertising. I understand that the US is a very polarized nation, so in that sense the US is a natural case. However, online political advertising is political speech and thus protected by the first amendment. As such, any effort to regulate targeted online advertising will be challenging at best. Looking at the four items that together measure attitude toward regulation, especially the first item: “Targeted political online advertising should be banned” seems to be measuring non-attitudes. I would invite the authors to construct a new scale with only items 2, 3, and 4 and subsequently run the relevant analyses once more. This scale will better capture US citizens’ attitudes toward regulation.

Response: We are grateful for this perspective. To address this concern, we re-ran our analysis and added the results to the paper. Please find the updated analysis in tables S9 and S10 in the appendix. Here, we find that the main effect persists and seems to get a little bigger with this specification. We still, however, report the full scale in the main body of the paper with reference to our pre-analysis plan. We also hope to follow the more ambitious legislative projects like the bill introduced by Rep. Anna Eshoo (1), which would ban online targeted political advertisement based on personal data on a federal level, in our research design to better understand how Americans think about a more radical approach to regulation. 

2. I think that knowledge about political targeting could be an important moderator for partisans’ estimates of the influence of targeted advertising on others. Why did the authors not include this variable and would the authors agree that this omission limits the interpretability of the findings?

Response: Dear Reviewer, thank you very much for pointing this out. We agree that knowledge about political targeting could be an important predictor of participants’ estimates of the influence of online targeted political advertising on others. In order to ensure that all participants had the same level of knowledge about political targeting we provided an explanatory text to participants in the beginning of the survey. After reading the text, participants had to indicate that they had read and understood the text. The text read as follows: 

„Please read the following information carefully. Targeted advertising is the practice of monitoring people’s online behavior and using the collected information to show people individually targeted advertisements. Online behavioral data can include web browsing data, search histories, media consumption data (e.g., videos watched), app use data, purchases, click-through responses to ads, and communication content, such as what people post on social networking sites. This online data is often combined with demographic data like age, gender and location. Political parties also use targeted advertising, for example, before presidential elections. Targeted political advertisement involves creating messages targeted at narrow categories of voters based on data analysis gathered from individuals' demographic characteristics and their online behavior. This enables political campaigns to send very specific messages to certain groups of potential voters. These messages are selected to be the most appealing to this group. Political actors use targeted advertising, for example, to reach voters who are likely to vote for them with messages that will influence them.”

With this text we intended to describe the basic mechanism behind targeted online advertisements, the data that is used for it, and potential applications in political campaigns. However, we agree that an additional measure of participants knowledge about targeted political advertisements would strengthen this assumption even more. Generally, the interpretation of our treatment manipulation should not be affected by any remaining differences in knowledge about targeted advertising, since participants were randomly assigned to the treatment or the control group after reading the text. We have added the following text to address this issue to the discussion section (p. 23, l. 601).: 

“Another limitation is that even though participants read an explanatory text about the functionalities and use cases of online targeted political advertising in the beginning of the survey, we could not rule out the possibility that participants had diverging levels of knowledge of the topic. While this does not limit the interpretation of our treatment manipulation, it could be an important predictor of participants' estimates of the influence of online targeted political advertising on others. Future studies should address this by measuring participants' knowledge after they have read an explanatory text.“

3. I am concerned about the phrasing of the treatment. Authors phrased the treatment as follows: “[…] The researchers found that the targeted political ads significantly increased the number of votes for the Republican party, but not for the Democratic party. Hence, targeted political ads influenced Republican voters, but did not influence Democratic voters.” First, this treatment text is not ‘truthfully’ informing participants about the effects in 2016. Essentially this text states that zero democratic voters were influenced? This can’t be the case. A small consequence Is that participants were, in fact, deceived. Second, and more importantly, why phrase the treatment in such an extreme, zero-sum, way? Why not state that targeting was thought to have benefitted republicans more than democrats? This treatment now is very artificial, unrealistic even. Which is a shame because there was a perfectly natural way to do it. As a result I wonder what the experimental effect really means? Of course republicans are not going to oppose something that gives them, and only them, an advantage. Such an advantage is worth setting aside, say, privacy concerns because it gives them back electoral victory and thus important policies. But in reality, the republican advantage, in 2016, was much more modest than described in the treatment. In a more realistic scenario republican partisan self-interest might have played a (much) smaller role.

Response: Thank you very much for this very important comment. The information we gave participants in the treatment was based on a working paper by Liberini et al. (2). In this working paper, the authors study the effect of exposure to online targeted political advertisements on Facebook during the 2016 U.S. presidential campaign on voter turnout and candidate choice. In order to do so, the authors exploit variations in prices of advertisements on Facebook to estimate the intensity of political campaigning for different audiences, specified by location, political affiliation and demographics. The idea is that given stable sizes of the audiences, more advertisements targeted to an audience should increase prices due to the demand shift. Hence, higher prices indicate higher campaign intensity. The authors then match this data with responses of 2,414 American voters from the 2016 American National Election Study (ANES) sampled once prior and once after the election regarding their Facebook use, turnout and candidate choice. The authors find that “targeted Facebook campaigning increased turnout among core Republican voters, but not among Democratic or Independent voters” (1: p. 5). They specifically state that “microtargeting was ineffective for Clinton, failing to boost turnout or sway voters in her favor” (1: p. 5). Thus, when designing our experiment, we based our treatment information on this working paper, aiming at changing the beliefs regarding the effect of online targeted political advertisements on opposing partisans of one group in our sample to be able to establish a causal relationship between partisan beliefs and support for regulation. Importantly, our search for scientific evidence of similar results of Democrats having been more influenced by online targeted political advertisements in some other election has not resulted in any meaningful findings. Therefore, as we did not want to deceive participants, we decided to manipulate only Republicans’ beliefs based on the evidence that we had at the time. After our survey had been in the field, a new version of the Liberini et al. (2) paper has been released in April, 2020. In this updated version, the authors conclude that being exposed to targeted online advertisement during the 2016 U.S. presidential campaign increased turnout for Trump supporters, and decreased turnout for Clinton supporters (3). Unfortunately, this information was not available to us when designing the study. We agree that the effect shown with our treatment manipulation might be smaller in the real world. Since our results come from a survey experiment, they are indeed somewhat artificial. Our aim was to examine whether altering participants’ self-interest considerations leads them to change their attitudes accordingly. With our experiment we are able to show that the mechanism of partisan self-interest motivating regulation preferences exists and that learning about the advantageous effects of the advertisements for their party leads Republicans to express lower support for a regulation that would oppose their partisan self-interest. We now discuss this aspect in the discussion section, stressing that partisan self-interest considerations might be smaller in the real world and that results of the 2018 working paper have been updated (p. 22, l. 583). 

4. Would it not be better to include independents as a control group? Or to compare the scores to a ‘balanced condition’, where both democrats as well as republicans benefit about equally from targeting?

Response: Thank you very much for this suggestion. We have not included Independents as a control group for two reasons. First, most Independents lean towards a party and are not completely neutral. Only about 7% of the American public did not favor a particular party in 2018 (4). Among Independents, a slightly higher share of 44% favored the Democratic party, while 34% favored the Republican Party in 2018 (4). Therefore, Independents would not yield a neutral control population. 

The second, and related, reason why we did not include Independents as a control group lies in the nature of the third-person effect (5). Specifically, our experiment was based on the fact that Democrats believe that Republicans are influenced more by online targeted political advertisements, and Republicans believe the opposite. This is analogous to findings showing that people believe that members of an out-group are more easily persuaded than members of their in-group, or themselves (5,6). The perception that out-group members are more influenced by persuasive communication than in-group members increases with social distance to the “others” (6–8). Due to increasing polarization between the parties, this distance is larger between Democrats and Republicans as compared to the distance between Independents and Democrats, or Independents and Republicans (9,10). Hence, the strength of the third person effect presumably varies between these groups and is stronger between Democrats and Republicans as opposed to Independents and Democrats, or Republicans. 

The average belief of Independents about whether Republicans or Democrats are more influenced by online targeted political advertisements should depend on the political leanings of the sample of Independents. In case of a majority of them leaning towards Democrats, their average belief should go in the direction of the belief of Democrats, but be somewhat less strong. If a majority of them would lean towards Republicans, the opposite would be true. Therefore, the effect of the treatment manipulation of informing Independents about Republicans having benefited more would depend on the composition of the sample of Independents. 

To answer your second question, in order to include a condition in which participants would be informed that Democrats benefited more from online targeted political advertisement, we would have needed scientific data to support this to avoid participant deception. However, as we mentioned above, at the time of designing the study, we could not find any evidence of Democrats heaving benefited more from the use of online political advertisements during some election. We did not want to deceive participants in our study, since we incentivized answers to our dependent variable by sending them to members of the United States Congress in aggregated and anonymized form. Therefore, we did not want the answers to be based on beliefs that might have been influenced by deception. Thus, we decided not to include such condition. Following your valuable feedback, we added a paragraph in the discussion section to address this topic (p. 22, l. 565). Thank you very much! 

5. Authors did never explicitly measure partisan self-interest. This makes it difficult to show the hypothesized mechanism. Why did they not measure this key concept?

Response: Thank you very much for addressing this important topic! We agree that we did not measure partisan self-interest explicitly. Instead, we propose that individuals are motivated by partisan self-interest if they believe the opposing party experiences an advantage from online targeted political advertisement and, as a consequence, try to mitigate this by supporting stricter regulation of these advertisements. The perceived advantage results from people believing that supporters of the opposing party are more easily influenced by mobilizing targeted advertisement from their respective party, which directly increases their chances of winning elections. The belief that supporters of the opposing party are more susceptible to online targeted political ads than supporters of the own party arises from the third-person effect (5). Hence, we argue that, within the context of our study, partisan self-interest is present when participants experience the third person-effect and act to mitigate its outcomes by supporting the regulation of online targeted political advertisement. 

In order to show that partisan self-interest plays a causal role in the support for regulation of online targeted political advertisement, we manipulate partisan self-interest in our experiment. To do so, we inform the treatment group about the advantageous effects of online targeted political advertising for Republicans by telling them that Republicans were more influenced by targeted advertisements than Democrats in the 2016 presidential election. This way, we manipulate Republicans’ beliefs about their co-partisans’ susceptibility to online targeted political advertisements in a way that shifts their partisan self-interest towards these ads being in their favor, helping them to win elections. Therefore, we are able to draw inferences about participants’ partisan self-interest by manipulating the treatment group’s beliefs. 

By asking participants about their beliefs a second time after measuring support for regulation, we are able to show that Republicans indeed shifted their beliefs towards thinking that Republicans, not Democrats, are more easily influenced by targeted advertisements. Hence, the treatment manipulation altered their perception of whether online targeted political advertisement would be in line with their partisan self-interest. 

Many other studies that examined partisan self-interest considerations in the support of election reforms used cross-sectional survey data to correlate participants’ political affiliation with their attitudes (11–16). This body of research concluded that public opinion towards electoral reforms often reflects party divisions, but could not establish causality. With our design, we follow several other scholars who have used similar approaches to study the causal role of partisan self-interest considerations in attitudes towards regulations or reforms concerning the electoral process. In a recent study about the influence of partisan self-interest on attitudes towards election reforms, partisan self-interest was manipulated using different treatment conditions (17). The treatment conditions were phrased such that the reforms were either in line with Democrats’ or Republicans’ partisan self-interest. Similarly, in a study about the role of partisan self-interest in attitudes towards same-day registration, participants were divided in three groups (18). The groups were either given a text that described same-day registration as being neutral, or favorable for Democrats or Republicans, thereby altering participants’ partisan self-interest considerations. An analogous approach was used in a study about support for voter identification, in which participants were assigned to treatments either presenting voter identification laws as increasing turnout for Democrats or Republicans (19). 

We have added more detailed descriptions of what we mean by partisan self-interest and how we manipulate it in the experiment to the introduction (p. 6, l. 120), the research design (p. 9, l. 200), the results (p.15, l. 381), and the discussion (p. 19, l. 471) sections. Thank you very much again for raising this point!

6. On p. 11 the authors state that perceived desirability of online targeted ads is low. But on a ten-point scale, perceived desirability is 4.66, sd = 2.01. I would not call this low.

Response: Thank you very much for pointing this out. We agree with you that this value is not low. We have rephrased this in the text (p. 12, l. 312; p. 18, l. 442). 

I believe this study makes an important contribution to the field of online political advertising research, especially when issues 1 and 3 are addressed.

Response: Dear Reviewer, thank you so much for your valuable and insightful suggestions and feedback. We have done our best to address your comments in this revised version of the manuscript. 

Comments from Reviewer 2

Manuscript PONE-D-20-33004 evaluates if Americans’ support for hypothetical regulations of microtargeting change when subjects are informed that microtargeting benefits Republicans while having no effect on Democrats. Results are not surprising based on the many published studies that find support for various (proposed) election laws shifts when partisans are informed that the proposed law disproportionately helps/hurt their party. In this experiment, the effects were only significant for Republicans – though, this is not clearly represented in the current wording of the abstract. I have a few concerns about the experimental components of the manuscript, as well as the article’s framings. Order is consistent with the manuscript’s layout and does not indicate priority.

Response: Dear Reviewer, thank you so much for your thoughtful feedback. Following your comment, we have adjusted the wording in our abstract to better reflect our findings. Specifically, we now state that only Republicans’ attitudes towards the regulation of online targeted political advertisement are partially motivated by partisan self-interest.

Further, while indeed multiple studies provide evidence that partisan self-interest considerations are associated with attitudes towards the election law, many of these studies are cross-sectional and, hence, do not establish causality (11–16). Several recent studies use experimental designs to measure causal effects of partisan self-interest on attitudes toward the election law (17–19). We follow this line of research to provide more experimental evidence of the causal relationship between partisans’ self-interest considerations and their opinions towards the election law. We now discuss these related studies in our discussion section (p. 20, l. 497).

Furthermore, the context of online targeted political advertising is relatively novel, with multiple stakeholders questioning current approaches to its regulations as well as attitudes of the constituents who drive this regulatory agenda. 

Microtargeting in the United States is very different than microtargeting in Europe due to the accessibility of publicly available electronic lists of registered voters. States are required by the Help America Vote Act of 2002 (HAVA) to compile and maintain these lists that include basic information (e.g. voter history, party registration, and race/ethnicity). This information is far more valuable for campaigns than data they can purchase or acquire elsewhere. The authors do not seem to understand the distinctions between microtargeting in the United States and elsewhere. And they fail to cite any scholars of US microtargeting (e.g. Hersch, Hillygus, Endres, Nickerson, Rogers, Panagopoulos).

Response: Dear Reviewer, thank you very much for pointing out the specifics of political targeting in the United States and for referring to these important scholars. To make it clearer that the focus of our study lies specifically on online targeted political advertisements, we now more clearly differentiate between “traditional” targeted political advertisements and online targeted political advertisement. In the introductory text that all participants read in the beginning of the study, we referred to online targeted political advertisement as well. 

Following your suggestions, we have included the findings of the researchers that were provided and have revised the introduction accordingly. In the introduction, we now mention a widespread use of large amounts of voter data for targeting in political campaigns that has become common since the implementation of the Help America Vote Act of 2002 (20–23). We further point out what is novel about online targeted political advertising in terms of the even larger amount of personal data being used, the increasingly advanced personalization techniques, and cost-effective ways to reach voters (24). We have also added more background information on legislation that has been introduced that would obligate stricter funding disclosure (25) or even ban online targeted political advertisements based on personal data (1). 

The study’s motivation feels like a straw man argument. The authors note on page 3, “a heated public debate calling for stricter regulations has accompanied the emergence of such ads.” However, none of these citations reference a “heated public debate” about regulating microtargeting in the United States, and I am not aware of a debate or proposed legislation in any US state or nationally. The merits of microtargeting are more frequently debated outside the US than inside. After all, the United States Congress paved the way for political microtargeting through their HAVA legislation (see Hersh 2015). The authors appear to be conflating debates about political misinformation, dark money in politics, and false statements on social media with political microtargeting. Further, the authors are somewhat cavalier with their references throughout the manuscript.

Response: Thank you very much for this very helpful comment. Instead of speaking about a heated public debate, we now base our argument on different actors who have raised concerns regarding the use of personal data for online targeted political advertising. We now refer to scholars who critique the use of personal data and the lack of disclosure and transparency (24,26–29). We also mention a private initiative that seeks to raise awareness about personal data use for political targeting (30) and we have added more recent evidence of the public’s negative attitude towards the use of personal data for online targeted political advertising (31). In addition to this, we cite two federal level legislations that were recently introduced and that aim for increasing disclosure obligations (25) or even banning targeted political advertisements on platforms (1). Further, based on your comment, we have revised all references of our manuscript. In the introduction, we have deleted those that referred to the harms of online targeted political advertising also in terms of misinformation and foreign interference. 

I do not consider the treatment, as described in the text, to be truthful. Though, the exact treatment language is not quoted in the text. The authors note, “we truthfully informed a randomly selected sample of participants that the Republican party benefited more than the Democratic party from the use of targeted political advertising in the 2016 presidential election” (page 6). I am not aware of any legitimate, peer reviewed publication that support this claim. It is possible, that Republicans benefitted more than the Democrats in terms of vote choice / persuasion, fundraising, (de)mobilization, but one CANNOT simply assume that the party of the winning candidate benefitted more than the party of the losing candidate. Later in the paper the treatment is described slightly differently as, “Participants in the treatment group were informed that controlling for the number of ads people saw, targeted political advertising on Facebook significantly increased voter turnout for the Republicans in the 2016 presidential election, while having no effect on Democrats ” (Page 8). The working paper cited to support this claim does not support it. In fact, it argues that targeted advertising had a negative effect on the turnout of Democrats.

Response: Thank you very much for this comment. We agree that it cannot simply be assumed that the winning candidate benefited more from the use of online targeted political advertisements. In our treatment, we informed participants that these advertisements “significantly increased the number of votes for the Republican party, but not for the Democratic party” and that they “influenced Republican voters, but did not influence Democratic voters”. We have added the instructions to the appendix. They are also available, together with the data and analysis files, at our data repository: https://osf.io/tynp7/?view_only=ccb6dcbfbb5a44ba985b572f959fb011. As noted in our response to Reviewer 1, the information we gave participants in the treatment was based on a working paper by Liberini et al. (2). In this working paper, the authors study the effect of exposure to online targeted political advertisements on Facebook during the 2016 U.S. presidential campaign on voter turnout and candidate choice. In order to do so, the authors exploit variations in prices of advertisements on Facebook to estimate the intensity of political campaigning for different audiences, specified by location, political affiliation and demographics. The idea is that given stable sizes of the audiences, more advertisements targeted to an audience should increase prices due to the demand shift. Hence, higher prices indicate higher campaign intensity. The authors then match this data with responses of 2,414 American voters from the 2016 American National Election Study (ANES) sampled once prior and once after the election regarding their Facebook use, turnout and candidate choice. The authors find that “targeted Facebook campaigning increased turnout among core Republican voters, but not among Democratic or Independent voters” (1: p. 5). They specifically state that “microtargeting was ineffective for Clinton, failing to boost turnout or sway voters in her favor” (1: p. 5). Thus, when designing our experiment, we based our treatment information on this working paper, aiming at changing the beliefs regarding the effect of online targeted political advertisements on opposing partisans of one group in our sample to be able to establish a causal relationship between partisan beliefs and support for regulation. Importantly, our search for scientific evidence of similar results of Democrats having been more influenced by online targeted political advertisements in some other election has not resulted in any meaningful findings. Therefore, as we did not want to deceive participants, we decided to manipulate only Republicans’ beliefs based on the evidence that we had at the time. After our survey had been in the field, a new version of the Liberini et al. (2) paper has been released in April, 2020. In this updated version, the authors conclude that being exposed to targeted online advertisement during the 2016 U.S. presidential campaign increased turnout for Trump supporters, and decreased turnout for Clinton supporters (3). Unfortunately, this information was not available to us when designing the study. In the discussion, we now report that results of the 2018 working paper have been updated (p. 23, l. 589) and we make it clear that the treatment text was written based on the results available at the time (p. 22, l. 581.).

More information is needed about the survey. Was there any quota sampling? If so, which variables were used? What was the time interval between each survey wave? What was the attrition rate? Some aspects/concepts are incorrectly described (e.g. a response scale from "not at all" to "to a very great extent" is NOT a Likert scale). Do you have a citations for the description of L2 as “the largest voting tracking service in the United States.” How did L2 and/or Dynata identify Democrats and Republican? Is this based on self-reports from an earlier survey? Party registration (not all states have party registration)? Votes in primary elections (most registered voters do not vote in primaries)?

Response: Dear Reviewer, thank you very much for raising these important questions. Quotas were applied on age, gender, region, and party affiliation. Following your comment, we have added this information in the sample description (p. 11, l. 262). Participants completed the survey within one session, no time passed between the different parts of the survey. We have made this clearer in the experimental design section (p.8, l. 173). We have further re-checked the information on L2. The information on being the largest such provider was based on self-description as provided on the website of L2 and we have not been able to independently verify this information. However, we found a description as “one of the largest” in Cappelen et al. (32). Therefore, we dropped this description. We also added additional information on L2’s methodology in the sample characteristics section, stating that respondents’ party affiliation was partially verified by their actual voting behavior and partially derived from other known attributes about the participants (p. 10, l. 257). Thank you very much for pointing out the wrong labelling of the scale, we have adapted the wording in the experimental design section (p. 8, l. 191). 

The "manipulation check", “all subjects were then asked to make an estimation of the number of interactions (likes, shares, comments) that social media campaigns on Facebook of both Republicans and Democrats received relative to each other prior to the midterm elections in 2018” does not appear to directly evaluate whether the treatment group read and/or processed the information in the treatment.

Response: We kindly thank you for this remark. We agree that we did not ask participants directly about their beliefs regarding the influence of targeted political advertisements on partisans in the 2016 election. With this question, we aimed at presenting participants with a similar, but not identical context as in the treatment. We chose the 2018 elections because this way participants had to infer from information they learned about the 2016 elections to an election in the future. Thereby, we could test if participants generalized from the treatment text to Republicans being more susceptible to online targeted political advertisements in general. This is important because of our dependent variable, which asks for attitudes for regulation of targeted advertisement. If participants thought Republicans were only more easily influenced in the 2016 election, they would not use this information for forming their support for regulation of online targeted political advertisements. However, we agree with you that this measure is not a strict manipulation check. Based on your feedback, we have rephrased and reflected this in the text (p.7, 144; p. 16, 358; as well as captions of S12 Fig.; S13 Fig.; S14 Fig.).

Finally, all control variables should be collected pre-treatment (See Gerber and Green 2012; Montgomery, Nyhan, Torres 2016; and many others), but it appears many were collected in wave 3 after the treatment was delivered (wave 2)

Response: Thank you very much for pointing that out. In our results section, we report the result of a Welch t-test which compares support for regulation between Republicans in the treatment and Republicans in the control group to report a result that is not influenced by the inclusion of control variables: “With Republicans, we found significantly lower support for stricter regulation of online targeted political advertising in the treatment than in the control group (two-sided Welch t-test, t(776) = 2.08, Cohen's d = 0.15, p = 0.04).” (p. 16, l. 389.). However, we have to admit that the discussion of the regression results in table 3 might raise concerns over the control variables having influenced the treatment effect. In response, we have deleted table 3 to make it clear that our main treatment effect is robust and not potentially influenced by the inclusion of control variables that were measured after the treatment was administered. For the same reason, we excluded the S10 table in the appendix. 

Further, it is important to emphasize that the correlational results that are reported in tables 1 and 2 are not influenced by the treatment as they are measured on respondents who did not receive the treatment information. These results show a correlation between privacy concerns and support for stricter regulation. 

Dear Reviewer, thank you very much for your insightful suggestions and valuable feedback. We have done our best to address your comments in this revised version of the manuscript. 

References:

1. Banning Microtargeted Political Ads Act, H.R.7014, 116th Congress. 2020. 

2. Liberini F, Redoano M, Russo A, Cuevas A, Cuevas R. Politics in the Facebook Era. Online Working Paper Series No 389, Centre for Competitive Advantage in the Global Economy, The University of Warwick. 2018;72. 

3. Liberini F, Russo A, Cuevas Á, Cuevas R, others. Politics in the Facebook Era-Evidence from the 2016 US Presidential Elections. CESifo Working Paper No 8235. 2020; 

4. Pew Research. Political Independents: Who They Are, What They Think. Pew Research Center - U.S. Politics & Policy. 2019. Available from: https://www.pewresearch.org/politics/2019/03/14/political-independents-who-they-are-what-they-think/

5. Davison WP. The Third-Person Effect in Communication. Public Opinion Quarterly. 1983;47(1):1. 

6. Perloff RM. Third-Person Effect Research 1983-1992: A Review and Synthesis. International Journal of Public Opinion Research. 1993;5(2):167–84. 

7. White HA. Considering Interacting Factors in the Third-Person Effect: Argument Strength and Social Distance. Journalism & Mass Communication Quarterly. 1997;74(3):557–64. 

8. Jang SM, Kim JK. Third person effects of fake news: Fake news regulation and media literacy interventions. Computers in Human Behavior. 2018;80:295–302. 

9. Iyengar S, Lelkes Y, Levendusky M, Malhotra N, Westwood SJ. The Origins and Consequences of Affective Polarization in the United States. Annual Review of Political Science. 2019;22(1):129–46. 

10. Bordalo P, Coffman K, Gennaioli N, Shleifer A. Stereotypes. The Quarterly Journal of Economics. 2016 Nov 1;131(4):1753–94. 

11. Gronke P, Hicks WD, McKee SC, Stewart III C, Dunham J. Voter ID laws: A view from the public. Social Science Quarterly. 2019;100(1):215–32. 

12. Wenzel JP, Bowler S, Lanoue DJ. Citizen opinion and constitutional choices: The case of the UK. Political Behavior. 2000;22(3):241–65. 

13. Bowler S, Donovan T, Karp JA. Why politicians like electoral institutions: Self-interest, values, or ideology? The journal of politics. 2006;68(2):434–46. 

14. Tolbert CJ, Smith DA, Green JC. Strategic voting and legislative redistricting reform: district and statewide representational winners and losers. Political Research Quarterly. 2009;62(1):92–109. 

15. Alvarez RM, Hall TE, Levin I, Stewart III C. Voter opinions about election reform: Do they support making voting more convenient? Election Law Journal. 2011;10(2):73–87. 

16. Stewart III C, Ansolabehere S, Persily N. Revisiting public opinion on voter identification and voter fraud in an era of increasing partisan polarization. Stan L Rev. 2016;68:1455. 

17. Biggers DR. Does partisan self-interest dictate support for election reform? Experimental evidence on the willingness of citizens to alter the costs of voting for electoral gain. Political Behavior. 2019;41(4):1025–46. 

18. McCarthy D. Partisanship vs. Principle: Understanding Public Opinion on Same-Day Registration. Public Opinion Quarterly. 2019;83(3):568–83. 

19. Kane JV. Why Can’t We Agree on ID? Partisanship, Perceptions of Fraud, and Public Support for Voter Identification Laws. Public Opinion Quarterly. 2017;81(4):943–55. 

20. Hillygus DS, Shields TG. The persuadable voter: Wedge issues in presidential campaigns. Princeton University Press; 2008. 

21. Nickerson DW, Rogers T. Political Campaigns and Big Data. Journal of Economic Perspectives. 2014;28(2):51–74. 

22. Hersh ED. Hacking the Electorate: How Campaigns Perceive Voters. Cambridge: Cambridge University Press; 2015 

23. Endres K, Kelly KJ. Does microtargeting matter? Campaign contact strategies and young voters. Journal of Elections, Public Opinion and Parties. 2018 Jan 2;28(1):1–18. 

24. Chester J, Montgomery KC. The digital commercialisation of US politics — 2020 and beyond. Internet Policy Review. 2019;8(4). 

25. Honest Ads Act, S.1989, 115th Congress. 2017. 

26. Burkell J, Regan PM. Voter preferences, voter manipulation, voter analytics: policy options for less surveillance and more autonomy. Internet Policy Review. 2019;8(4). 

27. Rubinstein I. Voter Privacy in the Age of Big Data. SSRN Journal. 2014 

28. Boerman SC, Kruikemeier S, Borgesius FJZ. Online Behavioral Advertising: A Literature Review and Research Agenda. Journal of Advertising. 2017 Jul 3;46(3):363–76. 

29. Wood AK, Ravel AM. Fool Me Once: Regulating Fake News and Other Online Advertising. S Cal L Rev. 2017;91:1223. 

30. ProPublica. What We Learned From Collecting 100,000 Targeted Facebook Ads. 2018. Available from: https://www.propublica.org/article/facebook-political-ad-collector-targeted-ads-what-we-learned

31. Gallup. In U.S., Most Oppose Micro-Targeting in Online Political Ads. 2020. Available from: https://news.gallup.com/opinion/gallup/286490/oppose-micro-targeting-online-political-ads.aspx

32. Cappelen AW, Haaland IK, Tungodden B. Beliefs about behavioral responses to taxation. Working Paper; 2018.

---

## [Decision Letter · Decision Letter 1]

8 Apr 2021

Partisan self-interest is an important driver for people's support for the regulation of targeted political advertising

PONE-D-20-33004R1

Dear Dr. Baum,

We’re pleased to inform you that your manuscript has been judged scientifically suitable for publication and will be formally accepted for publication once it meets all outstanding technical requirements.

Kind regards,

Natalie J. Shook

Academic Editor

PLOS ONE

Additional Editor Comments (optional):

Reviewers' comments:

Reviewer's Responses to Questions

**Comments to the Author**

1. If the authors have adequately addressed your comments raised in a previous round of review and you feel that this manuscript is now acceptable for publication, you may indicate that here to bypass the “Comments to the Author” section, enter your conflict of interest statement in the “Confidential to Editor” section, and submit your "Accept" recommendation.

Reviewer #1: All comments have been addressed

Reviewer #2: (No Response)

2. Is the manuscript technically sound, and do the data support the conclusions?

Reviewer #1: Yes

Reviewer #2: Yes

3. Has the statistical analysis been performed appropriately and rigorously? 

Reviewer #1: Yes

Reviewer #2: Yes

4. Have the authors made all data underlying the findings in their manuscript fully available?

Reviewer #1: (No Response)

Reviewer #2: Yes

5. Is the manuscript presented in an intelligible fashion and written in standard English?

Reviewer #1: Yes

Reviewer #2: Yes

6. Review Comments to the Author

Reviewer #1: Thank you for your elaborate and convincing rebuttal letter, and thank you for your contribution to the targeting literature.

Reviewer #2: I appreciate the authors’ response to the feedback provided by myself and the other reviewer. I agree the manuscript has improved overall.

I recognize the authors believed they were providing factual information (based on the cited working draft) when they conducted the survey. Now that they know otherwise, they should consider contacting the survey respondents assigned to the treatment group to notify them about the deception used in this study. The survey vendor (Dynata) should know which members of their panel participated in this survey, and, as such, it is possible to notify experimental subjects about the use of deception in this study. Further, the authors continue to use phrases such as “truthfully informing” in the text. At minimum, they should refrain from describing the treatment as truthful/accurate/honest. The content of the treatment can be presented to readers without claiming it is factual.

I agree with first reviewer that it is an overreach for the authors to claim they manipulated partisan self-interests. Instead, they randomized which subjects they exposed to information stating that targeted campaign communications benefitted one party and not the other. This information may or may not have influenced partisan self-interest.

The null effects for Democrats in the experiment is somewhat surprising. Generally, and based on the published experimental studies in this domain, you would expect Democrats and Republicans to move in opposite directions when provided with information that an aspect of elections benefits one party and not the other. After all, a campaign tactic that helps only one party mobilize their base has a net negative effect for the other party.

The exploratory analyses testing for heterogeneous treatment effects based on self-placement on a liberal-conservative scale is interesting. I am surprised the authors did not look at strength of party identification, which seems like a more natural choice.

The streamlined introduction is greatly improved. However, the last paragraph (of the introduction) is confusing (especially, the 2nd sentence).

7. PLOS authors have the option to publish the peer review history of their article (what does this mean?). If published, this will include your full peer review and any attached files.

Reviewer #1: No

Reviewer #2: No

---

## [Editor Report · Acceptance letter]

16 Apr 2021

PONE-D-20-33004R1 

Partisan self-interest is an important driver for people’s support for the regulation of targeted political advertising 

Dear Dr. Baum:

I'm pleased to inform you that your manuscript has been deemed suitable for publication in PLOS ONE. Congratulations! Your manuscript is now with our production department. 

Kind regards, 

on behalf of

Dr. Natalie J. Shook 

Academic Editor

PLOS ONE